# Targeting the NLRP3 Inflammasome in Glaucoma

**DOI:** 10.3390/biom11081239

**Published:** 2021-08-19

**Authors:** Sophie Coyle, Mohammed Naeem Khan, Melody Chemaly, Breedge Callaghan, Chelsey Doyle, Colin E. Willoughby, Sarah D. Atkinson, Meredith Gregory-Ksander, Victoria McGilligan

**Affiliations:** 1Northern Ireland Centre for Stratified Medicine, Ulster University, Londonderry BT47 6SB, UK; Coyle-S43@ulster.ac.uk (S.C.); Khan-M8@ulster.ac.uk (M.N.K.); s.atkinson@ulster.ac.uk (S.D.A.); 2Department of Molecular Medicine and Surgery, Karolinska Institute, SE-171 76 Solna, Sweden; melody.chemaly@ki.se; 3Centre for Molecular Biosciences, Biomedical Sciences Research Institute, Ulster University, Coleraine BT52 1SA, UK; b.conwell@ulster.ac.uk (B.C.); doyle-C29@ulster.ac.uk (C.D.); c.willoughby@ulster.ac.uk (C.E.W.); 4Department of Ophthalmology, Schepens Eye Research Institute, Massachusetts Eye & Ear Infirmary and Harvard Medical School, Boston, MA 02114, USA; Meredith_Gregory@MEEI.HARVARD.EDU

**Keywords:** NLRP3 inflammasome, glaucoma, RGC (retinal ganglion cells), inflammation

## Abstract

Glaucoma is a group of optic neuropathies characterised by the degeneration of retinal ganglion cells, resulting in damage to the optic nerve head (ONH) and loss of vision in one or both eyes. Increased intraocular pressure (IOP) is one of the major aetiological risk factors in glaucoma, and is currently the only modifiable risk factor. However, 30–40% of glaucoma patients do not present with elevated IOP and still proceed to lose vision. The pathophysiology of glaucoma is therefore not completely understood, and there is a need for the development of IOP-independent neuroprotective therapies to preserve vision. Neuroinflammation has been shown to play a key role in glaucoma and, specifically, the NLRP3 inflammasome, a key driver of inflammation, has recently been implicated. The NLRP3 inflammasome is expressed in the eye and its activation is reported in pre-clinical studies of glaucoma. Activation of the NLRP3 inflammasome results in IL-1β processing. This pro inflammatory cytokine is elevated in the blood of glaucoma patients and is believed to drive neurotoxic inflammation, resulting in axon degeneration and the death of retinal ganglion cells (RGCs). This review discusses glaucoma as an inflammatory disease and evaluates targeting the NLRP3 inflammasome as a therapeutic strategy. A hypothetical mechanism for the action of the NLRP3 inflammasome in glaucoma is presented.

## 1. Introduction

Glaucoma is a neurodegenerative disease and the leading cause of irreversible blindness worldwide. Glaucoma affects more than 70 million people, 10% of which are bilaterally blind [1,2]. The prevalence of glaucoma is estimated to increase to 111.8 million by 2040, which can be attributed to an aging population [3]. Glaucoma is characterised by degeneration of the retinal ganglion cell (RGC) axons resulting in damage or remodelling of the optic nerve head (ONH), as evidenced by the characteristic clinical sign of optic disc cupping [4]. These clinical manifestations lead to disruption of the visual pathway and vision loss in one or both eyes [5]. The most common type of glaucoma is primary open-angle glaucoma (POAG)—referred to as “glaucoma” in this review. POAG is characterised by diminished outflow of aqueous humour (AH) despite an unobstructed or open irideocorneal angle [6]. Many patients with glaucoma present with elevated intraocular pressure (IOP > 21 mmHg); however, this is not a requirement for the diagnosis of glaucoma, with 30–40% presenting with normal IOP [5].

The pathophysiology of glaucoma is not well understood and, currently, IOP is the only modifiable risk factor in the disease [7]. The goal of interventions for glaucoma are to lower IOP to a level that may prevent further damage to the ONH and therefore vision loss [5,6,7,8]. Interventions include topical medications that can be applied to the ocular surface, oral medications, laser treatment, and surgery to regulate AH outflow and production to lower IOP [1]. However, approximately 30–40% of POAG patients exhibit a normal IOP, indicating that elevated IOP is not the sole causative factor of glaucoma [9]. In fact a number of risk factors, such as genetics, age, and lifestyle, trigger common pathological endpoints resulting in glaucomatous optic neuropathy [10].

Impaired axonal transport in the RGCs, observed in both animal and human studies, is reported as a potential mechanism of damage in glaucoma [11]. Ischemia, restriction of blood to the bodies tissues, has also been implicated in the pathogenesis of glaucoma, with reduced ONH blood flow reported in glaucoma patients [12]. Excitotoxicity, which results in neuron damage, has been an area of interest in the pathophysiology of glaucoma; however, the data in this area are contradictory [5,13]. Moreover, there is an emerging body of evidence to support the role of inflammation in glaucoma pathogenesis [14,15].

## 2. Inflammation in Glaucoma

A number of recent studies have investigated the role of inflammation in glaucoma [16,17]. In experimental glaucoma, several studies reveal a significant induction of inflammatory genes in the ONH and retina in the early stages of glaucoma [18,19,20]. Inflammation in glaucoma primarily occurs in the retina and ONH [21]. Common triggers in glaucoma, such as vascular, mechanical, and immune triggers, all lead to astrocyte and microglial reactivity, neurotrophic factor deprivation, and oxidative stress [10]. Such cascades are believed to result in axonal damage in the ONH, which is the primary site of injury in glaucoma [10,22]. In human and experimental models of glaucoma, activated astrocytes [23,24] and activated microglia [25,26] are detected in the ONH, and coincide with the increased expression of proinflammatory cytokines, such as IL-1β and TNFα, and neurotoxic mediators, such as nitric oxide (NO), reactive oxygen species (ROS), and glutamate [26,27]. Inflammation and oxidative stress co-exist in glaucoma, as inflammation appears to increase the amount of oxidative stress and vice versa, creating a chronic state of inflammation and oxidative stress [28]. ONH astrocytes in glaucoma also overexpress cell adhesion proteins, which may promote the migration of immune cells to the site of ONH damage, further amplifying the immune response [29].

The retina has a self-defence system consisting of microglia, astrocytes, and Müller cells [30]. Chronic oxidative stress appears to drive inflammation through the activation of a para-inflammatory response in the retina [28], and is believed to be a result of aging [31]. Microglia in the aging retina appear to undergo morphological changes and also increase in number [32,33]. Para-inflammation involves low levels of inflammatory activation, which can occur in the microglia and can contribute to damage of the retina if dysregulated (Figure 1) [31]. A number of inflammatory cytokines involved in ageing are also found in the glaucomatous eye [34,35,36,37,38]. An increase in inflammatory cytokines IFN-γ, IL-6, IL-4, IL-10, and IL-1β resulted in a reduction of brn3a+ RGC cells in a glaucoma mouse model, and is thought to be related to microglial activation [39]. There is also evidence of complement activation in the glaucomatous retina [40]. The severity of RGC degeneration in glaucoma models can be correlated with microglial activation [41]. However, it is clear that axon degeneration and RGC death results from a system of complex interactions in glaucoma of many different cells and mediators [10].

Blocking activation of microglia in an experimental glaucoma model reduced RGC death, further strengthening the hypothesis of glial activation and inflammation playing a vital role in glaucoma pathogenesis [25,42]. Targeting inflammation or immune cells as a therapeutic option in glaucoma appears promising. Several novel anti-inflammatory inhibitors have been assessed in different animal models of glaucoma, targeting different inflammatory markers to improve glaucoma disease pathogenesis (Table 1). These inhibitors alter several physical and pathophysiological measures (see Table 1). Drawbacks are associated with some; however, for example, oryzanol failed to control IOP in an acute animal model of glaucoma [43], and lutein is ineffective at reducing increased levels of TNFα in chronic hypoxia [44]. However, the inhibition of specific inflammatory pathways may prove more efficacious than the current tested non-specific inflammatory inhibitors [45,46]. Many anti-inflammatory therapies that specifically target pro-inflammatory cytokines often experience challenges in human trials, such as increasing the risk of fatal infections [47,48].

Therefore, there is a need to develop novel but safe anti-inflammatory therapies [49,50]. Recently, the nucleotide-binding oligomerization domain, Leucine rich Repeat and Pyrin domain containing protein 3 (NLRP3) inflammasome, has gained attention as a potential key orchestrator of inflammation in the aetiology of glaucoma, and may be an attractive therapeutic target [21,51].

**Table 1 biomolecules-11-01239-t001:** Anti-inflammatory inhibitors tested in several in-vivo models of glaucoma.

Inhibitor of Inflammatory Markers	Inflammatory Markers	Glaucoma Model	In-Vivo Findings	References
*γ*-Oryzanol	TNF-*α* and IL-6	Subconjunctival injection of phenol in rabbit	Reduces IOP in a chronic glaucoma model by inhibiting the induction of TNF-α and IL-6, and provides protection against glaucoma	[43]
Fas inhibitor, ONL1204	Caspase-8, TNF-α, IL-1β, IL-6, and IL-18	Intracameral injection of microbeads in C57BL/6J mice	No effect on IOP. Prevents RGC death and axon degradation. Reduces microglial activation and inhibits induction of inflammatory cytokines and chemokines.	[52]
Myricetin	IL-1α, IL-1β, IL-6, and TNF-α	Injection of hyaluronic acid into the anterior chamber of the eye in Dawley rats	Lowers IOP level in animals and reduces inflammatory marker levels in in vitro experiments.	[46]
Lutein (hydroxycarotenoid)	TNF-α and IL-1β	Mouse model of retinal ischemia	Modulates the overexpression of GFAP in in vivo models of retinal ischemia and inhibits overactivation of NF-κB, IL-1β, and Cox-2 in Müller cells.	[44]
Puerarin	IL-1β, IL-17A, and TNF-α	Neovascular glaucoma in C57BL/6 mice	Puerarin reduces high levels of IL-1β, IL-17A, and TNF-α in animal models of glaucoma. It also maintains reactive oxygen species, superoxide dismutase and malondialdehyde, NOS, and inducible NOS and NF-κB to an optimum level.	[45]

IOP—intraocular pressure; RGCs—retinal ganglion cells; NOS—neuronal nitric oxide synthase; GFAP—glial fibrillary acidic protein; IOP—intraocular pressure; RGC—retinal ganglion cell; NF-κβ—nuclear factor-kappa beta; IL—interluekin; TNF-α—tumour necrosis factor-alpha; Cox-2—cyclooxygenase 2; NOS—nitric oxide synthases.

## 3. The NLRP3 Inflammasome

Pattern recognition receptors (PRRs) recognise danger signals, such as, pathogen associated molecular patterns (PAMPs) and danger associated molecular patterns (DAMPs) [53]. Nod like receptors (NLRs) are a type of PRR with a nucleotide-binding and oligomerisation domain, and act as receptors in the cytoplasm [50]. NLRs are categorised into four subgroups based on their N-terminal domain, with the NLRP group containing a pyrin domain [54]. NLRP3 is one member of the NLRP family, of which there are 14 members, all of which are involved in the formation of inflammasomes [55]. NLRP3 is the most well characterised member of the inflammasome family [56], and is an important regulator of inflammatory diseases and plays a key role in the innate immune system [57].

The NLRP3 inflammasome (NOD-, LRR-, and pyrin domain-containing protein 3) is an intracellular, multiprotein signalling complex implicated in a plethora of inflammatory diseases [55]. These integral elements include the sensor NLRP3 protein, an adaptor protein called adaptor molecule apoptosis-associated speck-like protein containing a CARD (ASC) and procaspase-1 [58]. The sensor NLRP3 protein can be triggered by PAMPs, DAMPs, and a range of diverse external stimuli such as infection and injury. NLRP3 inflammasome activation is a tightly regulated system and requires two signals for activation, as depicted in Figure 2. The priming signal results in an increased expression of NLRP3, pro-IL-1β, and pro-IL-18, as the nuclear factor-kappa beta (NF-κβ) pathway is activated in response to a stimulus [58]. The priming step of inflammasome activation is tightly controlled by a series of different post translational modifications of NLRP3, ASC, and caspase-1, including ubiquitination, phosphorylation, and sumoylation [59].

Triggering or activation is the second signal needed for inflammasome activation, and involves the oligomerisation of NLRP3 in its inactive form with procaspase-1 and ASC. Generally the activation step of inflammasome oligomerisation is a result of potassium efflux from the cell [59]. This results in the cleavage of procaspase-1 into caspase-1 and, as a result, pro-IL-1β and pro-IL-18 into active IL-1ß and IL-18, which are subsequently secreted from the cell [60,61]. As well as producing pro-inflammatory cytokines, active caspase-1 also results in the cleavage of gasdermin-D (GSDMD), a pore forming protein, resulting in pyroptosis, a form of cell death [62]. In the chronic diseases driven by inflammation, such as type 2 diabetes and cryopyrin-associated periodic syndromes (CAPS), the NLRP3 inflammasome is dysregulated, resulting in the uncontrolled release of pro-inflammatory cytokine IL-1β, which drives inflammation in such diseases [63,64].

## 4. IL-1 Signaling in the Eye

Interleukin 1 receptor 1 (IL-1R1) mediates interleukin 1 (IL-1) (IL-1α and IL-1β) signaling. Despite many studies alluding to the activation of microglia by IL-1, there is conflicting evidence in the literature that suggests that microglia do not express IL-1R1 [65]. However, IL-1R1 is clearly expressed in endothelial cells, astrocytes, and neurons in the eye [66]. IL-1α and IL-1β have been detected in various parts of the eye, including in tear fluid and corneal epithelial cells [67]. In the brain, IL-1 stimulates endothelial IL-1R1, which then produces factors that drive microglial inflammatory gene expression [68]. This process may also happen in the retina, as IL-1-stimulated inflammatory cytokine expression was largely abolished after the depletion of microglia, but the restoration of IL-1R1 on endothelial, but not microglial, cells restored IL-1 induced inflammatory gene expression [66]. Retinal pigment epithelial cells and the trabecular meshwork of humans also express IL-1R1 [69]. Fibroblasts are also known to highly express IL-1R1, and may contribute to microglial stimulation. Fibrosis is a known pathological response in the lamina cribrosa in glaucoma [70,71], and it is likely that there is a cross talk between the lamina cribrosa and retinal cells, which ultimately leads to the activation of microglial cells and subsequent RGC death [65,72].

## 5. NLRP3 in the Eye

The NLRP3 inflammasome is constitutively expressed in various parts of the eye including the retinal pigment epithelium and ONH astrocytes in both human and mice, indicating the importance of this mediator in the defense system of the eye [73,74,75]. NLRP3 is also expressed in many other cells of the eye, including the ONH, retinal microglia, Müller cells [76,77,78], and astrocytes [78,79]. NLRP3 expression is not limited to the retina and ONH, and is found throughout the eye in the conjunctiva, trabecular meshwork, retinal pigment epithelium, and corneal epithelial cells in disease states [80,81,82,83,84].

## 6. The NLRP3 Inflammasome in Glaucoma

Dysregulation of the NLRP3 inflammasome has been implicated in several neurodegenerative diseases, including Alzheimer’s disease and multiple sclerosis [85]. Increased levels of IL-1β mRNA and protein have also been observed in the blood of glaucoma patients compared with controls, suggesting activation of the NLRP3 inflammasome in glaucoma [35]. Furthermore, activation of the NLRP3 inflammasome has been associated with the induction of IL-1β and death of RGCs in mouse models of acute glaucoma via optic nerve (ON) crush. [64,86]. In an inducible mouse model of glaucoma, the use of fluorescent reporter mice to track inflammasome activation demonstrated that NLRP3 inflammasome activation occurs early in the ONH, following IOP elevation, and coincides with the induction of pro-inflammatory cytokines and Iba1+, a microglia marker, immune cells in the ONH [87]. This upregulation of inflammatory genes occurs as early as 7 days post IOP elevation in the ONH and before the induction of inflammatory genes occurs in the retina [79]. With the use of knockout mice lacking various components of the NLRP3 inflammasome, we can conclude that the NLRP3 inflammasome activation is required for RGC death [64,79,88]. These studies support the hypothesis that “danger” signals in the eye, IOP elevation being one, activates the NLRP3 inflammasome pathway in the glial cells of the ONH and retina, resulting in neurotoxic inflammation, axon degeneration, and subsequent death of RGCs in glaucoma models.

## 7. Proposed Mechanism for the Role of NLRP3 in Glaucoma

Chronic inflammation from unknown causes, ageing, or genetics may be initiating factors [1], and may directly lead to the activation of glial cells or, alternatively, may lead to damage to the trabecular meshwork [89]. NLRP3 is expressed and activated in the trabecular meshwork as a result of oxidative stress, leading to elevated IOP, further exacerbating the initiation of inflammation [80,90].

Initial injury may also be due to increased IOP resulting in mechanical stress on various cells such as fibroblasts in the sclera [91,92], and subsequently or alternatively, lamina cribrosa astrocytes of the ONH. Fibroblasts and astrocytes are known to express high levels of IL-1R1, and the NLRLP3 inflammasome is known to be activated in these cell types [65,78,93,94,95].

A number of risk factors trigger common pathological endpoints resulting in glaucomatous optic neuropathy [10]. In glaucoma, the ON glial cells become activated, which in turn damage the axons leaving the eye and further trigger inflammatory cell recruitment to the injury sites [79,96]; ONH astrocytes constitutively express NLRP3 and injury to the ONH may also activate the NLRP3 inflammasome pathway in these cells [79,96]. As Figure 2 depicts, the NF-κβ pathway is activated by DAMPs or injury to the ON, resulting in transcriptional upregulation of pro-IL-1β and NLRP3 [58]. A second signal, generally resulting in potassium efflux from the cell, results in the oligomerisation of NLRP3, ASC, and pro-caspase-1 to form the inflammasome complex [30,59]. We hypothesise that extracellular ATP binding to the ligand-gated ion channel purinergic type 2 receptor 7 (p2x7r) or ROS produced by oxidative stress as a result of ageing are some methods of inflammasome activation in glaucoma that may be independent of IOP [30,76,97]. The ATP concentration is higher in the aqueous humour of glaucoma patients compared with the controls, and increased levels of ATP in the glaucomatous eye are thought to be released from cells damaged or stressed by elevated IOP [98,99,100]. However, sheer stress and cell swelling in tissues can also increase extracellular ATP in the absence of elevated IOP [101,102]. Moreover, it has been proposed that ROS produced from reactive astrocytes contribute to early axonal damage within the ONH in glaucoma [103]. Amyloid beta, a known inflammasome activator in the brain microglia, is also increased in the glaucomatous eye compared with normal eyes, and may result in retinal glial cell activation [104,105]. TGF-β, a profibrotic cytokine is found in the aqueous humour and ONH astrocytes in glaucoma patients. TGF-β is involved in damage to the ONH in glaucoma [106] and TGF-β can induce activation of the NLRP3 inflammasome, which has also been demonstrated to be involved in fibrosis [107]. Upon inflammasome activation, pro-caspase-1 is cleaved into its active form caspase-1, which can in turn cleave pro-IL-1β into its active form IL-1β, which is then secreted from the cell [61]. In glaucoma, IL-1β was found to be one of the first inflammatory cytokines upregulated in the ONH, and is a powerful stimulus for immune cell recruitment, further supporting the role of neurotoxic inflammation as a significant contributor to the neurodegenerative process of glaucoma [108,109]. Therefore, pharmacologically targeting NLRP3 may serve as a neuroprotective therapy to prevent the progression of glaucoma.

## 8. NLRP3 Inflammasome as a Target for Therapy in Glaucoma

There is an obvious need for the development of new medications for glaucoma, given that current treatments work to lower IOP only. However, these treatments are not successful in all patients [110,111]. NLRP3 may be a promising IOP independent target for the treatment of glaucoma [112]. Several studies demonstrate that the inhibition of NLRP3 activation significantly inhibits the death of RGCs in experimental models of retinal ischemia/reperfusion injury (acute glaucoma) [113,114,115,116]. Blockade of the P2x7r, which can act as signal 2 in inflammasome activation with A438079, a P2x7r inhibitor, also attenuates RGC death by inhibiting inflammasome activation. Additionally in an in-vitro model, ATP, which binds to the P2x7r, was shown to induce inflammation in the retina by activating the NLRP3 inflammasome [76]. High-mobility group box 1 (HMGB1), which is often released from damaged cells, is known to be released upon rapid IOP elevation and is also a known DAMP that activates Toll Like Receptor 4 (TLR-4) to activate the NF-kB pathway. In an acute glaucoma mouse model, the release of HMGB1 resulted in increased levels of NLRP3 and IL-1B. Inhibition of HMGB1 resulted in reduced NLRP3 and IL-1B levels, which also reduced RGC death and glaucoma severity [117].

However, in the more clinically relevant microbead-induced mouse model of glaucoma, where IOP was elevated, treatment with a commercially available NLRP3-specific inhibitor (MCC950) [118] was also shown to prevent axon degeneration and death of RGCs [119]. In this microbead-induced model of glaucoma, 4 weeks of elevated IOP resulted in a 25–30% loss of RGCs, which was significantly attenuated following a three-times a week treatment with intraperitoneal MCC950. Mice treated with MCC950 displayed axon and RGC densities equal to that of the non-glaucoma controls. MCC950 injection, however, had no effect on IOP elevation in the treated mice [119]. Currently, this is the only study that has evaluated a NLRP3-specific inhibitor in an inducible model of glaucoma [119]. Collectively, these studies provide evidence that NLRP3 inhibition may be a novel therapeutic strategy to protect RGCs and prevent axon degeneration in glaucoma.

MCC950, a small molecule drug, however, does present with some drawbacks, including the very short half-life of MCC950 of only 3 h [120]. This means the drug must be administered regularly either systemically or by multi-intravitreal injections, both of which are not desirable for diseases of the eye. Other limitations of small molecule inhibitors include that they are often not fully characterised, and many are not completely specific to their targets [121,122]. Therefore, there has been interest in recent years in the development of biologics, such as antibodies for the treatment of many chronic conditions, as they have revolutionised the treatment and modification of many diseases including some cancers and autoimmune rheumatic diseases [121,122]. Biologics are well known to have a longer half-life, allowing for longer dosing intervals because of their larger size, which is an important factor in the treatment of glaucoma [122,123]; they are also recycled by the body and have a high affinity and potency. The major advantage of biologic molecules for treatment is the specificity of their mechanisms of actions preventing off target effects [122]. Financial cost in the long run is also an advantage of biologic therapies, as they are expected to deliver a better economic return than small molecules [121]. The use and approval of biologics in medicine is increasing each year, targeting extracellular or cell membrane proteins [123]. However, the delivery mechanisms of antibodies into cells to target intracellular molecules is a major obstacle [123]. The current inhibitors of NLRP3 and IL-1β are outlined in Table 2. It is important to note that, generally, biologics targeting the NLRP3/IL-1β pathway target secreted IL-1β directly, like Canakinumab to suppress its inflammatory responses, which poses a serious risk of fatal infection, as IL-1β cannot be produced in response to an infection [49]. Targeting NLRP3 to supress IL-1β production means IL-1β can still be produced by other pathways if the body encounters an acute infection [124]. Anakinra is another biologic that targets the NLRP3 pathway binds to IL-1R1 to inhibit IL-1β binding to IL-1R1, as this can further activate the NLRP3 inflammasome [125]. There is a number of small molecule inhibitors that target NLRP3 to suppress its effects or inhibit NLRP3 inflammasome oligomerisation [114,126,127,128,129].

## 9. Conclusions

Elevated IOP is the only modifiable risk factor in glaucoma, and IOP lowering therapies are unsuccessful in many patients. There is a need for IOP independent therapies to treat or slow glaucoma progression. There is significant evidence to support the role of inflammation in the pathogenesis of glaucoma. In particular, the NLRP3 inflammasome pathway in glial cells of the ONH and retina appears to play a critical role in axon degeneration and death of RGCs in glaucoma. The NLRP3 inflammasome therefore appears to be a strong target for therapy development in glaucoma. A biologic therapy with a long dosing interval would be desirable to avoid multiple dosing. Additional studies are now required to further investigate current and novel inhibitors of the NLRP3 inflammasome pathway for glaucoma treatment.

## Figures and Tables

**Figure 1 biomolecules-11-01239-f001:**
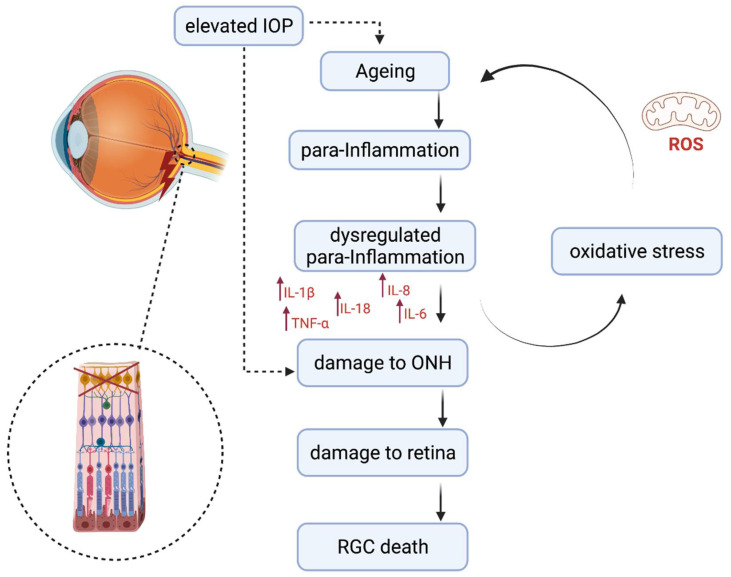
Aging and para-inflammation contributes to retinal ganglion cell death. Ageing in the retina due to ROS is thought to activate a para-inflammatory response, which can become dysregulated and result in damage to the ONH and retina and subsequently RGC death ROS—reactive oxygen species; RG—retinal ganglion cells; ONH—optic nerve head.

**Figure 2 biomolecules-11-01239-f002:**
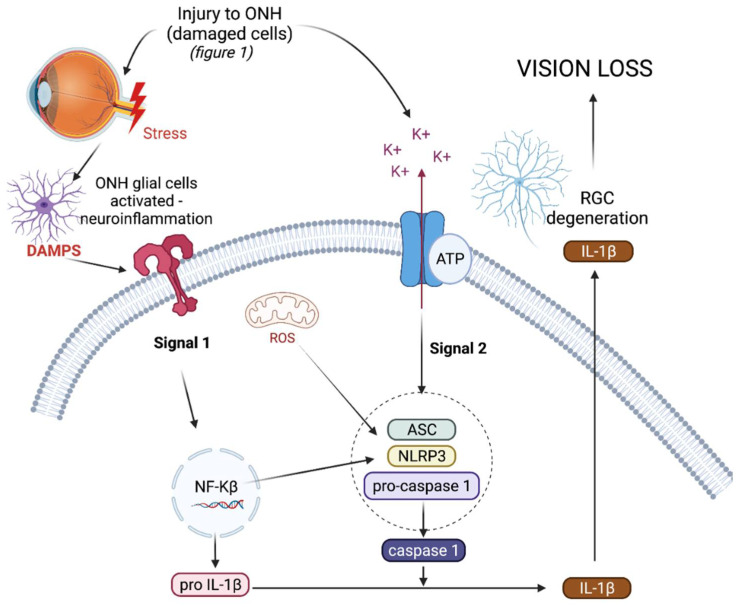
Hypothetical mechanism for the role of NLRP3 in glaucoma. Activation of the NLRP3 inflammasome requires two signals for activation. Signal 1 results in transcriptional upregulation of Pro-IL-1β and NLRP3 protein via the NF-κβ pathway. Stress to the ONH is thought to activate the NF-κβ pathway. Signal 2 results in the oligomerisation of ASC, NLRP3, and pro-caspase-1 to form the NLRP3 inflammasome. NLRP3 inflammasome oligomerisation results in cleavage of pro-caspase-1 into active caspase-1, which can in turn cleave pro-IL-1β into IL-1β, which can leave the cell to drive inflammation. ROS or extracellular ATP are proposed to result in inflammasome oligomerisation in glaucoma. ONH—optic nerve head; ROS—reactive oxygen species; ATP—adenosine triphosphate; RGC—retinal ganglion cell; DAMPS—danger associated molecular patterns; IL-1β—interluekin-1beta; ASC—apoptosis-associated speck-like protein containing a caspase-recruitment domain; NLRP3—NLR family pyrin domain containing 3; NF-κβ—nuclear factor-kappa beta.

**Table 2 biomolecules-11-01239-t002:** Novel IL-1β and NLRP3 inhibitors.

Inhibitors	Type	Mechanism of Action	Reference
Canakinumab	Biologic	Inhibits binding of IL-1β to IL-1R1	[49]
Anakinra	Biologic	Binds to IL-1 receptor to inhibit IL-1β binding	[125]
MCC950	Small molecule	Inhibits caspase-1 activation by binding to NLRP3	[126]
CY-09	Small molecule	Inhibits NLRP3 ATPase activity by binding to the ATP-binding motif of the NACHT domain	[127]
Tranilast	Small molecule	Suppresses NLRP3 assembly by binding to NACHT domain	[128]
IC-100	Biologic	Inhibits adaptor ASC	[130]
Argablin	Small molecule	Inhibits NLRP3 activation (specific mechanism not known)	[129]
Cholchicine	Small molecule	Inhibits expression of components of NLRP3	[113]

IL-1β—interleukin-1beta; IL-1R1—interleukin 1 receptor 1; NLRP3—NLR family pyrin domain containing 3; ATP—adenosine triphosphate; NACHT—a central nucleotide-binding and oligomerization domain; ASC—Apoptosis-associated speck-like protein containing a caspase-recruitment domain.

## Data Availability

Not applicable.

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
