# Peer review of "Targeting the NLRP3 Inflammasome in Glaucoma"

_biomolecules, 2021, doi:10.3390/biom11081239_

Round 1
Reviewer 1 Report
In this review the authors take the focus in a very interesting field, the search of new treatment therapeutic in the neuroprotection way in glaucoma, considering that many cases of glaucoma do not course with hypertension and therefore, cannot be treated with hypotensive classic treatment. The authors provide an overview of the mechanism of action of the NLRP3 inflammasome and how this factor may constitute a target in the treatment of glaucoma. The manuscript is clearly and correctly written and easy to understand. The authors should make some modifications to improve the quality of the manuscript, specified bellow:
Minor comments:
- The authors have included a section entitled "Inflammation in glaucoma". In this section the authors should include some recent work related to the state of para-inflammation in the aging retina and the release of cytokines by microglia and macroglia and their role in glaucoma progression.
Ramírez, A. I., Fernández-Albarral, J. A., de Hoz, R., López-Cuenca, I., Salobrar-García, E., Rojas, P., ... & Ramírez, J. M. (2020). Microglial changes in the early aging stage in a healthy retina and an experimental glaucoma model. Progress in Brain Research, 256(1), 125-149.
Fernández-Albarral, J. A., Salazar, J. J., de Hoz, R., Marco, E. M., Martín-Sánchez, B., Flores-Salguero, E., ... & Ramírez, A. I. (2021). Retinal Molecular Changes Are Associated with Neuroinflammation and Loss of RGCs in an Experimental Model of Glaucoma. International Journal of Molecular Sciences, 22(4), 2066.
Ramírez, A. I., de Hoz, R., Fernández-Albarral, J. A., Salobrar-Garcia, E., Rojas, B., Valiente-Soriano, F. J., ... & Salazar, J. J. (2020). Time course of bilateral microglial activation in a mouse model of laser-induced glaucoma. Scientific reports, 10(1), 1-17.
- In the abstract authors seems to have forget a word in the follow sentence: “The NLRP3 inflammasome is expressed in the…. and its activation is reported in pre-clinical studies of glaucoma” Please correct it.
- In table 1 please change TNFα by TNF-α in the second row
- In the section “Proposed mechanism for the role of NLRP3 in glaucoma” second paragraph, the follow sentence seems to be incomplete; Initial injury may also be due to increased IOP resulting in mechanical stress on various cells such as fibroblasts in the….. [88,89],
- In the section “NLRP3 inflammasome as a target for therapy in glaucoma” authors should briefly state what A438079 is.
- Authors should include a flow chart with the search criteria used in this review.
- The last paragraph after table 2 could be included as "Conclusion" to clarify that it is the conclusion of this work or start with "in conclusion".
- The authors should correct the references section; there are some references that do not have the same format as the rest, specifically: 75, 85, 86, 87, 88, 93, 115, 118.

Author Response
Dear Reviewer 1,
Please see attachment.
Kind regards,
Sophie

Reviewer 2 Report
The authors discuss glaucoma as an inflammatory disease, evaluating the NLRP3 inflammasome target NLRP3 as a therapeutic strategy.
However, they do a very short review (less than 3000 words, when the minimum is 4000). They make a summary of the literature, but I think they can improve the explanation related to the reviewed papers. For exmple, they do not explain in the text some of the papers mentioned in table 2 (121-123). Moreover, they do not explain which the P2x7 receptor is. They can improve this items to improve the manuscript.
Minor revisions
The authors should correctly include abbreviations throughout the manuscript, including tables and figure legends. For example, they wrote RGC directly, but should have written retinal ganglion cells the first time they name it in the manuscript. Other examples are IL-1, TFF, NOD, LRR, NLRP3, etc. As well as they write optic nerve head several times throughout the manuscript but they should write ONH. Please correct them.
The references are not correctly included. Some of them are in black and not underlined.
In the section “Proposed mechanism for the role of NLRP3 in glaucoma” there is a sentence cut off, just before the citations 88,89. Please correct them
The title of Table 1 is not on the same page where the table appears. Please correct this. Also, in Puerarin's "Inflammatory markers" they write (B) and (C) which are not explained.
I suggest you, to improve the appearance of both tables.
In Table 1: Il-6 and IL-17A are on different line. I suggest you to change the column width to correct it.
In table 2. The Reference column presents a large width, which is not necessary in my opinion. Correcting the appearance of the tables would greatly improve the appearance of the paper.
The last paragraph of the manuscript should have a corresponding title. Conclusions or concluding remarks. I suggest the authors to include it.
At the “back matter” of the manuscript, authors should include "author contribution" and "conflict of interest".
They should also review the references, since they do not have for number [113] the same format as the other references.
Author Response
Dear Reviewer 2,
Please see the attachment.
Kind regards,
Sophie

Round 2
Reviewer 2 Report
The suggested revisions have been made, so I recommend this revised version of the manuscript for publication, which is currently quite improved, although it does not reach 4000 words.
However, there are some points that need the Authors' attention to improve the final version and hence the impact of their research work:
Line 4 --> The authors have removed reference 19 in the text, but not in the reference list. They should make sure that the order of the references is correct.
Lines 9 and 11 --> Some references do not have the same format. In fact, they appear in black and not underlined. This is a minor aspect, but easy to correct.
Line 13 --> Reference number 30 is cited before 28 and 29. The order in which they are cited should be corrected.
Lines 172-176 --> I think the authors have mistakenly deleted the paragraph. Because they have corrected the number of references cited in this text, however they have crossed out the corresponding sentences. If this is not the case, they should correct the order of the deleted references in the text
Line 212 --> I believe that the current reference number 125, Netea et al, is not included in Table 2.
Tables 1 and 2 --> I suggest the authors write the abbreviations at the end of the table, in the table footnote, and not in the title. This gives a cleaner look to the information provided in the table.
Author Response
Dear Reviewer 2,
Please see attached.
Kind regards,
Sophie
